# Identification of Differentially Expressed Genes and Molecular Pathways Involved in Osteoclastogenesis Using RNA-seq

**DOI:** 10.3390/genes14040916

**Published:** 2023-04-14

**Authors:** Sarah Rashid, Scott G. Wilson, Kun Zhu, John P. Walsh, Jiake Xu, Benjamin H. Mullin

**Affiliations:** 1School of Biomedical Sciences, University of Western Australia, Perth, WA 6907, Australia; 2Department of Endocrinology & Diabetes, Sir Charles Gairdner Hospital, Nedlands, WA 6009, Australia; 3Department of Twin Research and Genetic Epidemiology, King’s College London, London SE1 7EH, UK; 4Medical School, University of Western Australia, Perth, WA 6907, Australia

**Keywords:** bone resorption, osteoclasts, osteoporosis, differential gene expression, RNA sequencing

## Abstract

Osteoporosis is a disease that is characterised by reduced bone mineral density (BMD) and can be exacerbated by the excessive bone resorption of osteoclasts (OCs). Bioinformatic methods, including functional enrichment and network analysis, can provide information about the underlying molecular mechanisms that participate in the progression of osteoporosis. In this study, we harvested human OC-like cells differentiated in culture and their precursor peripheral blood mononuclear cells (PBMCs) and characterised the transcriptome of the two cell types using RNA-sequencing in order to identify differentially expressed genes. Differential gene expression analysis was performed in RStudio using the edgeR package. Gene Ontology (GO) and Kyoto Encyclopedia of Genes and Genomes (KEGG) pathway analyses were performed to identify enriched GO terms and signalling pathways, with inter-connected regions characterised using protein–protein interaction analysis. In this study, we identified 3201 differentially expressed genes using a 5% false discovery rate; 1834 genes were upregulated, whereas 1367 genes were downregulated. We confirmed a significant upregulation of several well-established OC genes including *CTSK*, *DCSTAMP*, *ACP5*, *MMP9*, *ITGB3*, and *ATP6V0D2*. The GO analysis suggested that upregulated genes are involved in cell division, cell migration, and cell adhesion, while the KEGG pathway analysis highlighted oxidative phosphorylation, glycolysis and gluconeogenesis, lysosome, and focal adhesion pathways. This study provides new information about changes in gene expression and highlights key biological pathways involved in osteoclastogenesis.

## 1. Introduction

The structural integrity of the human skeleton is maintained by a constant renewal process known as bone remodelling, in which old bone is degraded and new bone is generated [1]. Osteoblasts and osteoclasts (OCs) are the two primary bone cell types that play a role in the remodelling process [2]. Osteoblasts are specialised cells of mesenchymal origin that participate in the synthesis of new bone. On the other hand, OCs are derived from monocyte/macrophage lineage cells which fuse to form multinuclear giant cells [3,4]. OC formation is regulated by two essential cytokines: receptor activator of NF-kB ligand (RANKL) and macrophage colony-stimulating factor (M-CSF) [3,5]. M-CSF is produced by marrow stromal cells and osteoblasts, whereas RANKL is a member of the TNF protein superfamily and is produced by T-cells, stromal cells, and osteoblasts [6]. Receptor activator of NF-kB (RANK) and colony-stimulating factor-1 receptor (CSF1R) are the respective receptors for RANKL and M-CSF and are expressed in osteoclast precursor cells. The binding of M-CSF to CSF1R is crucial for the proliferation and survival of osteoclast precursor cells, while the binding of RANKL to RANK generates signals that are important for osteoclast differentiation, resorptive function, and the survival of mature osteoclasts [7]. Excessive osteoclast activity can contribute to certain pathological bone disorders, such as Paget’s disease, bone metastasis, and osteoporosis [8].

Osteoporosis is a complicated bone disease with hormonal, genetic, and nutritional contributions. It is typically associated with reduced bone mineral density, abnormal bone microarchitecture, and decreased bone strength, increasing the propensity for fractures [9,10]. The disease is characterised by an imbalance between resorption and formation that affects the homeostasis of the bone, that is, there is more resorption than formation, leading to structural weakness [11,12]. Osteoporotic fractures are a leading cause of morbidity and mortality in the elderly. It has been estimated that roughly half of all women and a quarter of all men over the age of 50 will experience bone fragility fractures in their remaining lifetime [13]. Among all fractures, hip fracture is the most devastating, with a one-year mortality rate of 36% in men and 21% in women [14]. This results in a significant medical and socioeconomic burden for society, which will rise as the population ages.

Finding new potential targets for the treatment of osteoporosis will lead to improvements in the quality of life for elderly individuals and a reduction in the high costs associated with the disease. RNA sequencing (RNA-seq) is a well-established technique that allows the profiling of the whole transcriptome, thus permitting the identification of new targets, signalling pathways, alternative splicing, and changes in gene expression between different conditions [15]. RNA-seq is a widely used technology in skeletal biology research and represents a useful tool by which we can expand our understanding of the underlying mechanisms involved in osteoporosis [16].

OCs have been studied extensively due to their role in bone resorption [17,18] and are commonly targeted by antiresorptive therapies used to maintain skeletal integrity [19]. It has been established that peripheral blood mononuclear cells (PBMCs) can act as precursor cells for OCs [20], with numerous studies utilising these cells to investigate OC formation and its role in osteoporosis [21,22]. In this study, we differentiated OC-like cells in culture [23] and used RNA-seq to identify genes that are differentially expressed when compared to their precursor PBMCs. Bioinformatics analysis was performed to investigate the protein–protein interaction (PPI) and subnetwork modules, perform gene enrichment analysis, and to identify enriched signalling pathways to characterise the underlying mechanisms regulating OC differentiation.

## 2. Materials and Methods

### 2.1. Subject Recruitment and Generation of Osteoclast-like Cells

For differential gene expression (DGE) analysis, 8 female study participants aged 30–70 and of European ancestry were recruited. Exclusion criteria were used to ensure that the participants had no chronic medical conditions and did not use medications that affect osteoclastogenesis or osteoclastic bone resorption. All study participants provided written informed consent, and the study was approved by the Sir Charles Gairdner and Osborne Park Health Care Group Human Research Ethics Committee. PBMCs were collected by density gradient centrifugation from whole blood samples obtained from each subject according to well-established protocols in our laboratory [23,24,25,26]. In brief, blood tubes were centrifuged at 2200 rpm for 13 min before the buffy coats were collected and diluted with PBS, and then gently layered over Ficoll-Paque Premium (GE Healthcare, Chicago, IL, USA). Centrifugation was performed and the PBMC layers were collected. Washing and centrifugation were performed again, and the supernatant was removed. The resulting PBMC cell pellet, containing a mixture of primarily lymphocytes and monocytes, was re-suspended in 1 mL of complete α-MEM (supplemented with 10% foetal bovine serum and 1% penicillin/streptomycin) containing 25 ng/mL M-CSF. The cells were counted and each sample was used to seed 6 wells (2 sets of triplicates) of a 24-well cell culture plate with 1.5 × 10^6^ cells each. The cells were then incubated at 37 °C with 5% CO_2_ for two days, after which the medium from one set of triplicates was removed, along with the non-adherent lymphocytes, and the adherent cells (primarily monocytes) were harvested. For the other set of triplicates, the medium was removed, and the adherent cells were cultured with complete α-MEM containing 25 ng/mL M-CSF and 100 ng/mL RANKL to generate OC-like cells. After an additional 12 days in this medium formulation, the OC-like cells were harvested. The osteoclastic nature of the cultures was confirmed by staining for tartrate resistant acid phosphatase (TRAP), as previously described [25].

### 2.2. Nucleic Acid Extraction

Each batch of triplicate cell cultures was gently washed with PBS before nucleic acid extraction was performed. Genomic DNA and total RNA were harvested from each culture using the AllPrep DNA/RNA Mini Kit (QIAGEN, Hilden, Germany). The cell lysates from each set of triplicate cultures were combined into a single aliquot. DNase digestion was performed for the RNA fraction of each sample to ensure there was no DNA contamination. The quality of each RNA sample was assessed using an Agilent 2100 Bioanalyzer (Agilent Technologies, Santa Clara, CA, USA), with high-quality RNA obtained for the samples (RNA integrity numbers all ≥ 8.1).

### 2.3. Data Processing for Gene Expression

Gene expression quantification was performed on the RNA samples extracted from the PBMC and OC cultures by the Australian Genome Research Facility (AGRF) using 100 bp single-end RNA-Seq on an Illumina NovaSeq platform (Illumina, San Diego, CA, USA). The Illumina bcl2fastq 2.20.0.422 pipeline was used to produce primary sequence data, with sequence quality values across all bases assessed for the 16 samples. Cleaned reads were aligned against the Homo sapiens genome (build version hg38), and the STAR aligner v2.5.3a was used to map reads to genomic sequences. Raw counts were summarised at the gene level using the featureCounts v1.5.3 utility of the Subread package [27].

### 2.4. Differential Gene Expression Analysis

DGE analysis between the PBMC and OC groups was performed using the edgeR package in RStudio [28,29]. Genes with low expression levels were removed, and trimmed mean of the M values (TMM) normalisation of the data was performed. Differentially expressed genes (DEGs) were identified using the quasi-likelihood F-test with multiple testing corrections performed using the Benjamini–Hochberg method [30]. To determine DEGs, we used a 5% false-discovery rate (FDR) and log fold-change (logFC) threshold ≥ 1 or ≤−1. A volcano plot was generated using the ggplot2 package in RStudio and a heat map was created for the top 30 genes using the in-built heatmap function in RStudio. Box plots for specific genes were also generated using ggplot2, with *p*-values for these calculated using a *t*-test performed with the ggprism package in RStudio.

### 2.5. Functional Enrichment Analysis of DEGs

GO analysis was performed using the web-based tool, the Database for Annotation, Visualization and Integrated Discovery (DAVID, version 6.8) [31]. Gene annotations including biological process, molecular function, and cellular component were performed by submitting up- and downregulated DEGs separately into the online DAVID tool. The results from DAVID were imported into RStudio and the ggplot2 package was used to generate bubble plots. To perform the KEGG pathway analysis for up- and downregulated DEGs, the java-based program Gene Set Enrichment Analysis (GSEA) [32] was utilised. The ranked gene file was used as an input to run GSEAPreranked and the c2.cp.KEGG database with 186 gene sets was used as a reference set for pathway analysis. Normalised enrichment score (NES) and FDR for enriched pathways were represented in GSEA plots, and a 5% FDR was used throughout. GSEA results were also presented in the form of bar plots generated using the ggplot2 package in RStudio.

### 2.6. Protein–Protein Interaction (PPI) Network and Module Analysis

A web-based tool, the Search Tool for the Retrieval of Interacting Genes (STRING) database [33], was utilised to generate a PPI network for up- and downregulated DEGs separately, with an interaction score > 0.4. The STRING output was then analysed using the Cytoscape software (v3.9.0) [34]. Cluster analyses of the PPI network for both up- and downregulated DEGs were performed using the Cytoscape plugin Molecular Complex Detection (MCODE) [35]. To perform the cluster analysis, the following parameters were used: node score cut-off = 0.2, degree cut-off = 2, k-score = 2, and max. depth = 100. KEGG pathway analyses were performed for both up- and downregulated genes using the online tool DAVID, with FDR-corrected *p* < 0.05 considered significant.

## 3. Results

### 3.1. Differential Gene Expression Analysis between PBMC and OC Groups

After filtering, normalisation and quality control (QC) were applied; DGE analysis identified 3201 DEGs between the PBMC and OC groups, with 12,280 being non-significant. Of the 3201 genes, 1834 were upregulated and 1367 were downregulated in the OC compared to the PBMC group (Appendix A). A volcano plot for the DEGs in the OC group is displayed in Figure 1a, with the top 30 DEGs presented in a heat map (Figure 1b).

### 3.2. Analysis of Established OC Genes

The DGE results for several genes known to have a role in OC differentiation and/or function were checked in our data to provide confidence in our OC-like cell culture model. These included *cathepsin K* (*CTSK*), *dendrocyte expressed seven transmembrane protein* (*DCSTAMP*), *tartrate-resistant acid phosphatase type 5 (ACP5*), *matrix metalloproteinase 9* (*MMP9*), *ATPase H+ transporting V0 subunit d2* (*ATP6V0D2*), and *integrin subunit beta 3* (*ITGB3*). All of these genes were found to be significantly upregulated in the OC group with logFC ≥ 1 and *p* < 0.05 (Figure 2a–f). Furthermore, we checked the expression levels of the monocyte marker genes *C-C motif chemokine receptor 2* (*CCR2*)*, C-C motif chemokine receptor 5* (*CCR5*), and *selectin L* (*SELL/CD62L*) in the PBMC and OC groups. All three of these genes demonstrated significantly greater expression in the PBMC group compared to the OCs (Appendix A).

### 3.3. Enrichment Analysis of DEGs

Using the DAVID software, the top 10 enriched biological processes identified for the upregulated genes include positive regulation of MAPK cascade, cyclin-dependent protein serine/threonine kinase activity, lipid metabolic process, cholesterol biosynthetic process, cell migration, and cell adhesion (Figure 3a and Appendix A). The downregulated genes are primarily involved in the inflammatory response, immune response, chemokine-mediated signalling pathway, signal transduction, and cell surface receptor signalling pathway (Appendix A). The top 10 enriched molecular functions identified for upregulated genes include protein kinase binding, oxidoreductase activity, microtubule binding, integrin binding, and collagen binding (Figure 3b and Appendix A), whereas downregulated genes are associated with transmembrane signalling receptor activity, C-C-chemokine receptor activity, transcription factor activity, carbohydrate binding, signalling receptor activity, and integrin binding (Appendix A). We also identified the top 10 cellular component annotations for the upregulated genes, which include the plasma membrane, lysosomal lumen, integral component of the plasma membrane, extracellular exosome, endoplasmic reticulum membrane, cytoplasm, and cell surface (Figure 3c and Appendix A). The downregulated genes are enriched in the plasma membrane, cell surface, extracellular region, and receptor complex annotations (Appendix A).

The GSEA pathway analysis for upregulated genes suggested that they are enriched in a number of functional annotations (Figure 4a–d, Appendix A). Several of these annotations present as being related to osteoclast function, including focal adhesion, lysosome, and the metabolic pathways of oxidative phosphorylation and glycolysis and gluconeogenesis. This provides additional confidence in the OC-like nature of the differentiated cells. Downregulated genes are enriched in a number of annotations including cytokine–cytokine receptor interaction, hematopoietic cell lineage, T-cell receptor interaction, and Nod-like receptor PI3K- signalling pathways (Appendix A).

### 3.4. PPI Interaction Network Analysis

PPI network analysis identified the most inter-connected genes in both up- and downregulated DEG sets. The top 3 gene clusters in the upregulated DEGs are significant with MCODE scores of 57.7, 12.7, and 8 (Appendix A). The top three gene clusters identified for the downregulated DEGs are presented in Appendix A. KEGG analysis was performed for the genes identified in the upregulated gene cluster using the DAVID software (Appendix A). It was found that the genes in cluster 1 are associated with a number of pathways including the cell cycle, cellular senescence, and P53 signalling pathways. Cluster 2 genes are associated with steroid biosynthesis, valine, leucine, and isoleucine degradation, and metabolic pathways, whereas cluster 3 genes are enriched in the oxidative phosphorylation, phagosome, and metabolic pathways. Some of the biological pathways enriched for downregulated DEGs identified in the KEGG analysis (Appendix A) for cluster 1 include cytokine–cytokine receptor interaction, NF-kappa B signalling pathway, TNF signalling pathway, and IL-17 signalling pathway. Cluster 2 genes were enriched in cell adhesion molecules, hematopoietic cell lineage, and ECM–receptor interaction, whereas no significant pathways were identified for cluster 3 genes.

## 4. Discussion

In this study, we have performed DGE analysis between OC-like cells and their precursors, identifying over 3000 genes whose expression levels changed during osteoclastogenesis. Significant upregulation of several known OC genes was established in the OC group, such as *CTSK*, *DC-STAMP*, *ACP5*, *MMP9*, *ATPV0D2*, and *ITGB3*, which gives confidence in the OC-like nature of the cell culture model. In addition, several genes identified as upregulated in the OCs have genetic evidence for association with bone traits. One of these is *carboxypeptidase E* (*CPE*), a locus that we have previously reported as harbouring co-localised association signals for estimated BMD (eBMD) [36] and expression of the *CPE* gene in human OC-like cells [23]. CPE has a role in prohormone processing and vesicle transport for secretion [37]. It has been demonstrated that the expression of *Cpe* is upregulated in mouse OCs compared to precursor bone marrow-derived macrophages, with the authors suggesting that the gene may be an important modulator of OC differentiation induced by RANKL [38]. Our data in human cells support this idea. Another example is the *creatine kinase B* (*CKB*) gene, which encodes brain-type creatine kinase and has a role in ATP homeostasis. The *CKB* locus also harbours a genome-wide significant association signal for eBMD [36]. Previous studies have shown upregulation of the *Ckb* gene during osteoclastogenesis in mice, and that interfering with the expression of this gene results in suppressed osteoclastic bone resorption [39].

The KEGG analysis revealed that the focal adhesion annotation is enriched for genes upregulated during osteoclastogenesis. Focal adhesions (FAs) are large macromolecule protein complexes composed of extracellular matrix, integrins, cytoskeleton, and other proteins, and they influence various cellular processes such as proliferation, migration, differentiation, spreading, and apoptosis [40,41]. FAs act as a mechanical linkage connecting the cytoskeleton with the extracellular matrix, thereby acting as a signalling hub that can sense the external as well as the internal microenvironment, allowing cells to perform outside-in and inside-out signalling [42]. There is evidence to suggest that FAs may play a role in maintaining the integrity of the skeletal system [43,44]. It has been found that mature OCs form a specialized sealing zone in order to bind to the bone surface and perform their bone resorptive activity [45]. It has been suggested that integrins play a vital role in OC activity by promoting OC adhesion and regulating cytoskeletal organization, thus enabling bone resorption [46]. The integrin αVβ3 is the most prominent integrin found in OCs [47] and acts as a major adhesion receptor. This integrin adheres to bone matrix proteins such as osteopontin, vitronectin, and bone sialoprotein by recognizing their amino acid RGD (Arg-Gly-Asp) motif [48]. A deletion of the β3 integrin subunit gene in mice results in the formation of abnormal OCs with diminished capacity to resorb bone [46]. The intracellular domain of integrin αVβ3 in OCs activates various signalling molecules including Src [49], Syk [50], and Pyk2 [51] as well as cytoskeleton proteins [45,51]. It also activates Rac family GTPases [47,52] and the guanine nucleotide exchange factor, VAV3 [8], each of which has an essential role in the arrangement of the cytoskeleton.

During the process of bone resorption, mature OCs adhere tightly to the bone surface at the sealing zone, forming an isolated microenvironment between themselves and the bone surface. Protons and proteolytic enzymes are then secreted into this space, where they degrade the mineral and organic matrix components of the bone. Previous studies have highlighted the importance of metabolic reprogramming in OCs in order to meet the energy demands of this cell type [53]. OCs contain a large number of mitochondria which are thought to be crucial for their differentiation and bone resorptive function [54,55]. Both OC formation and bone resorption are highly energy-demanding metabolic processes, and OCs require large quantities of adenosine triphosphate (ATP) [55,56]. Energy generated through metabolic pathways in OCs is an emerging area of research [57,58]. In our analysis, we have identified oxidative phosphorylation as an enriched pathway for genes upregulated during osteoclastogenesis, which is important for the production of ATP [59]. Oxidative phosphorylation has been identified as a primary source of energy for OC formation, and the proper assembly of mitochondrial complex 1 is fundamental for oxidative phosphorylation and osteoclastogenesis [60]. In addition, RANKL/RANK signalling has been implicated in the expression of mitochondrial biogenesis factors and the regulation of OC formation through the activation of oxidative phosphorylation. It has been found that RANKL/RANK signalling stimulates the expression of MYC which leads to the expression of EERα. EERα is a mitochondrial biogenesis factor; therefore, MYC/EERα may help fuel OC formation via activation of oxidative phosphorylation [61]. Another enriched pathway we have identified in this study for genes upregulated in the OCs is glycolysis and gluconeogenesis. Glycolysis is the metabolic pathway that converts glucose into ATP and pyruvate, while gluconeogenesis is required for the synthesis of glucose [58]. Previous studies have reported the importance of glycolysis in OC differentiation and resorption. It has been found that during osteoclastogenesis, the uptake of glucose and expression of glycolytic genes is increased; therefore, the inhibition of the glycolysis pathway or the depletion of glucose in culture media causes impaired osteoclastogenesis [56]. Glucose is considered an essential nutrient for bone resorption as it enhances the attachment of mature OCs to the bone surface, thus indicating a role for glucose in OC resorptive function [62]. It has been shown that glycolysis increases in mature OCs, and the degradation activity on type 1 collagen was found to be increased when mature OCs were exposed to glucose-containing media [63]. Furthermore, it has been shown that glucose regulates the expression of vacuolar H(+)-ATPase in OCs, which is important for their resorptive activity [64].

Lysosomes are intracellular organelles with typically acidic contents and their main function is to degrade intracellular and extracellular material [65]. In this study, we identified the lysosomal pathway as enriched for genes differentially expressed during osteoclastogenesis. The importance of lysosomal proteins in OC function has been highlighted by previous studies [66,67]. Lysosomes play a critical role in bone resorption by regulating OC activity. It has been demonstrated that lysosomes create an acidic environment in the resorption lacuna as well as help to synthesise and secrete the proteases that are essential for the degradation of bone extracellular matrix [68]. It has been widely reported that acidification in resorption lacuna depends on the presence of different subunits of the vacuolar-type ATPase (V-ATPase) proton-driven pump [69]. We confirmed the significant upregulation of one of the better-characterised V-ATPase subunit genes, *ATP6V0D2*, in the OC-like cells. The expression of this gene has been found to be essential for early OC differentiation and extracellular acidification [70]. An association of the d2 and a3 subunits of the V-ATPase pump has been identified as important for bone resorption [71]. Another V-ATPase subunit gene, *ATP6V1C1*, is critical for lysosomal acidification in OCs; it is also found to co-localise with the a3 subunit of the V-ATPase pump, and is involved in F-actin ring formation during OC activation [72]. We also confirmed a significant upregulation of the key lysosomal protease genes *CTSK, ACP5*, and *MMP9* in the OC-like cells. Previous studies indicate a role for lysosomal proteases in the degradation of the organic components of bone, thus promoting bone resorption [66,67]. We have also identified the lysosomal pathway genes *SYT7* and *SNX10* as being upregulated in the OC-like cells, with previous studies documenting their involvement in exocytosis and trafficking of lysosomal vesicles, thus playing a vital role in bone resorption [73,74].

There is a potential limitation in this study regarding the use of RNA-seq for comparative analyses, the results from which would ideally require confirmation through quantitative PCR or quantitation of protein expression levels. However, contemporary RNA-seq is generally regarded as a robust procedure, and, in the context of the higher sequencing depth that is used, it generates more informational reads, which increases the statistical power to confidently detect differential expression among genes with lower expression levels, and typically does not require independent validation [75].

In conclusion, we have performed a DGE analysis of OC-like cells and their PBMC precursors. Over 3000 genes were identified as differentially expressed between the 2 cell types, including several genes with an established role in OC biology such as *CTSK*, *DCSTAMP*, *ACP5*, *MMP9*, *ATP6V0D2*, and *ITGB3*. Pathway enrichment analysis highlighted several key biological pathways enriched for differentially expressed genes, including several potentially linked with osteoclastic bone resorption such as focal adhesions and lysosomal proteins. Several pathways related to metabolic activity were also identified as enriched, including glycolysis, gluconeogenesis, and oxidative phosphorylation, potentially highlighting the high energy requirements of this cell type. The results from this study have identified some of the key genes and pathways that are involved in OC differentiation and activity, some of which may represent potential therapeutic targets for the treatment of bone disease.

## Figures and Tables

**Figure 1 genes-14-00916-f001:**
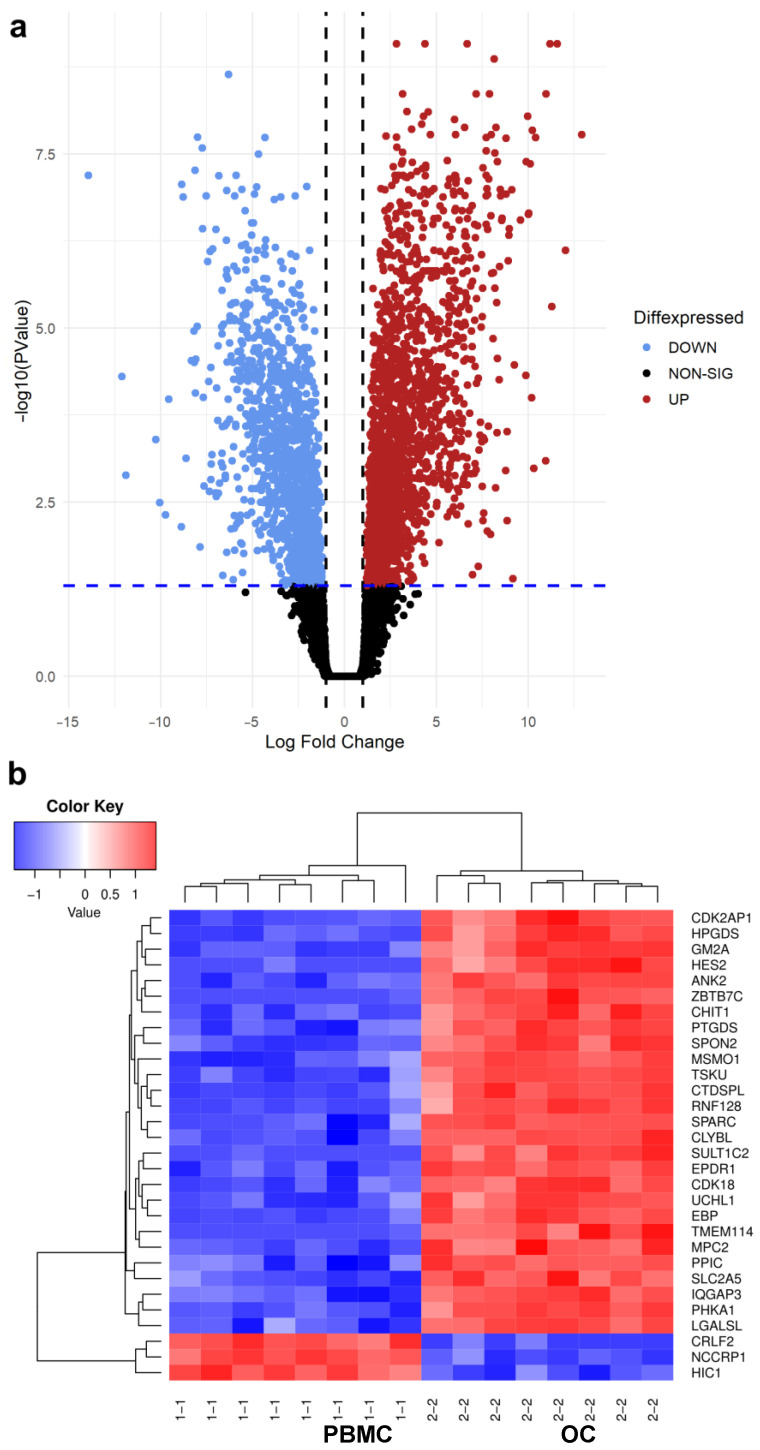
(**a**) Volcano plot illustrating the DEGs between the OC and PBMC groups. Genes significantly up- and downregulated in the OC group relative to the PBMC group are indicated using red and blue dots, respectively. (**b**) Heat map based on the hierarchical clustering for the top 30 differentially expressed genes identified between the PBMC and OC groups.

**Figure 2 genes-14-00916-f002:**
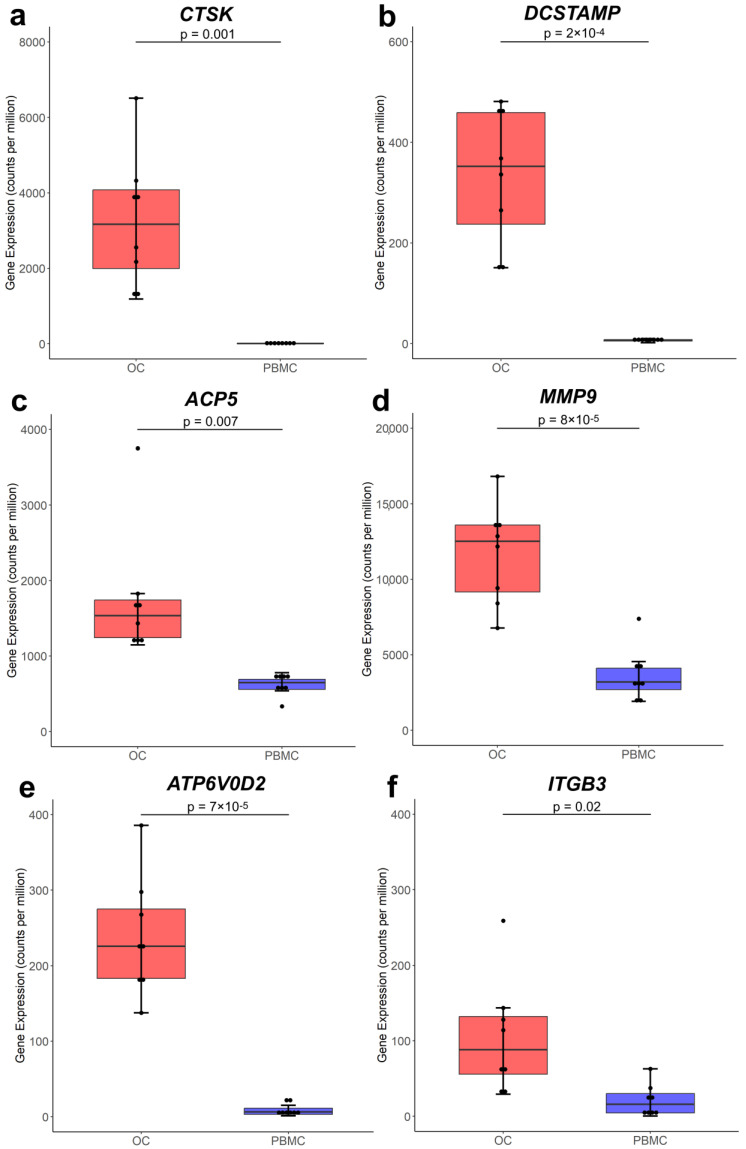
Box plots displaying the relative expression of established OC genes: (**a**) *CTSK*, (**b**) *DCSTAMP*, (**c**) *ACP5*, (**d**) *MMP9*, (**e**) *ATP6V0D2*, (**f**) *ITGB3*. The box plots are filled in red (left) and blue (right) representing the OC and PBMC groups, respectively. The y-axis indicates the expression of the genes in counts per million, whereas the x-axis denotes the OC and PBMC groups. A solid horizontal line in the middle of each box plot represents the median gene expression value. *p*-values in these plots were calculated using a *t*-test, with *p* < 0.05 considered statistically significant.

**Figure 3 genes-14-00916-f003:**
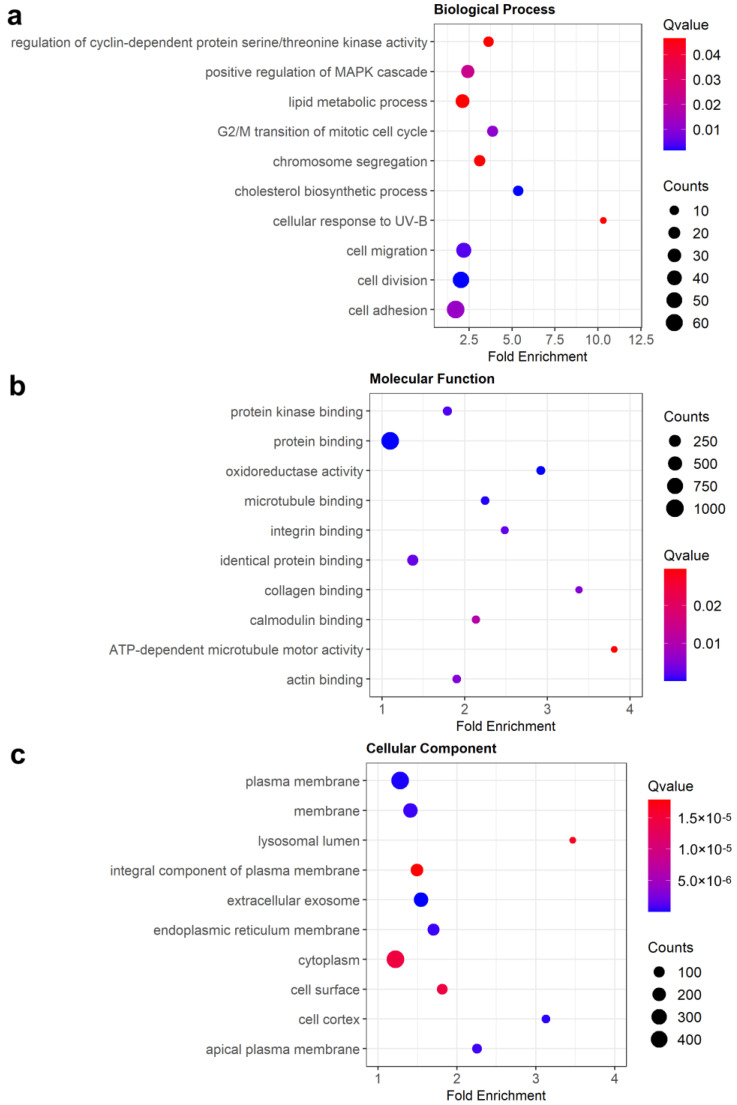
Bubble plots for GO analysis of upregulated DEGs. The top 10 enrichment terms from the GO analyses are plotted for (**a**) biological process, (**b**) molecular function, and (**c**) cellular component.

**Figure 4 genes-14-00916-f004:**
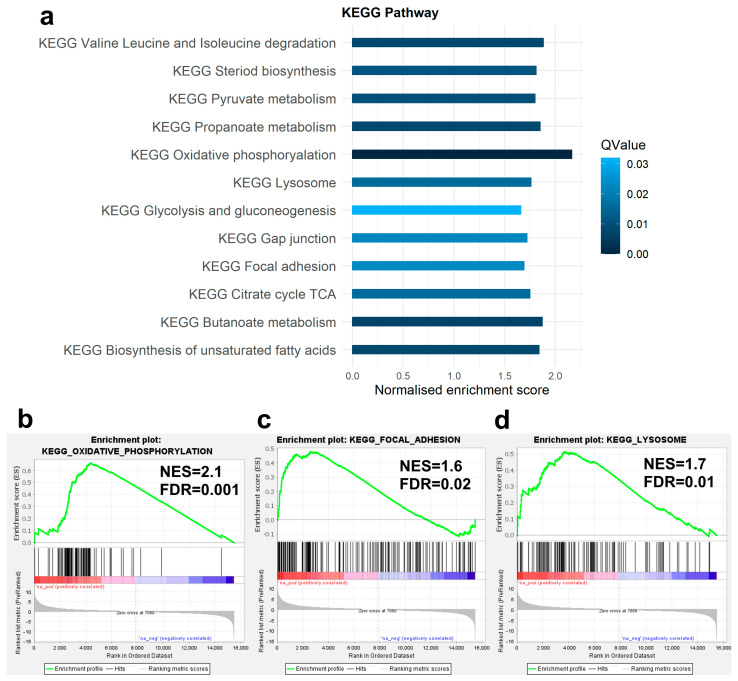
Results from the KEGG analysis performed using the GSEA software. (**a**) Bar plot displaying the results from the KEGG analysis. The GSEA plots (**b**–**d**) display gene sets that are enriched for upregulated genes in the OC group. Normalised enrichment score (NES) and false discovery rate (FDR) are presented.

## Data Availability

RNA sequencing data have been deposited in the GEO database (accession number GSE225974).

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
