# Peer review of "Identification of Differentially Expressed Genes and Molecular Pathways Involved in Osteoclastogenesis Using RNA-seq"

_genes, 2023, doi:10.3390/genes14040916_

Round 1

Reviewer 1 Report

This manuscript by Rashid et al compared PBMC and OC-like transcriptional profiles through RNA-seq analysis. The methods are detailly described. The data are interpreted. However, the results presented are much more like a preliminary bioinformatic analysis, and the knowledge we learn from the current presentation is pretty few. More work should be developed to improve the significance of the study.

Majors.

1. Figure 2, only shows OC-like high expressed genes, It should also include PBMC specific genes and validation by RT-qPCR.

2. Beyond those known genes, what's the new genes/pathways we can get from RNA-seq to indicate successful OC differentiation.

3. For better understanding of OC-like generation steps, more time-point should collect and repeat the analysis.

4. GSEA for those pathways with negative NES scores should be also included in the manuscript.

Minors

1. Lane 83, spelling errors for "Foetal bovine serum'.

Author Response

Reviewer 1

  • Figure 2 only shows OC-like high expressed genes. It should also include PBMC specific genes and validation by RT-qPCR.

Thank you for this suggestion. We have added 2 additional sentences to the “Analysis of Established OC Genes” section of the Results, and a new supplementary figure to the manuscript (Supplementary Figure S2) displaying expression levels for 3 monocyte-specific genes in the PBMC and OC cell cultures. It is not possible to perform RT-qPCR on these RNA samples as they were used up in the RNA-seq assays. However, recent studies have shown that RNA-seq can generally be considered robust enough to identify differentially expressed genes without requiring validation with RT-qPCR (Coenye, PMID 33665610).

  • Beyond those known genes, what's the new genes/pathways we can get from RNA-seq to indicate successful OC differentiation.

We have added some text to the “Enrichment Analysis of DEGs” section of the Results about new enriched pathways identified in the analysis that potentially indicate successful osteoclast differentiation. Staining for the osteoclast marker tartrate-resistant acid phosphatase (TRAP) has also been performed on these cells to help confirm successful differentiation. This has been added to the “Subject Recruitment and Generation of Osteoclast-like Cells” section of the Materials and Methods. We have published extensively using OC-like cells generated with this protocol (see Mullin et al. 2014, 2018, 2019, 2020) and have confirmed their osteoclastic nature further by establishing the capacity for resorption of bone and verifying expression levels of established osteoclast marker genes.

  • For better understanding of OC-like generation steps, more time-point should collect and repeat the analysis.

This type of longitudinal data collection is beyond the scope of this cross-sectional study. Such a study might be considered for the future; however, additional funding would need to be secured as the RNA-seq costs would likely be quite high considering the number of assays that would need to be run for multiple time points. Furthermore, it is unclear whether such additional studies would provide much new information, since the cross-sectional study we performed would be expected to highlight the main driving gene expression differences under the well-established conditions investigators in the field use for osteoclast generation.

  • GSEA for those pathways with negative NES scores should be also included in the manuscript.

We have described the pathways that we discovered in our analysis as enriched for down-regulated genes on page 12, lines 4-6. Table S4 in the supplementary file contains all the information about the up- and down-regulated pathways we identified in the GSEA analysis, along with their normalised enrichment scores.

  • Lane 83, spelling errors for “Foetal bovine serum”.

Thank you, this has been corrected.

Reviewer 2 Report

The manuscript by Rashid et al. comprehensively analyzed gene expression differences between osteoclast-like (OC-like) and peripheral blood mononuclear cells (PBMCs) based on bulk RNA-seq. They found the differentially expressed genes and their GO enrichment categories. The major findings and bioinformatic analysis are convincing. However, one fundamental question, whether the OC-like cell was successfully differentiated or not, needs further experimental proof. Following are the points of criticism.

1. In the experiment design, the author used two factors, M-CSF and RANKL, to generate OC-like cells for 12 days of cultures. Is there any literature to support this cultural method? If not, the experiment is needed to prove the success of OC-like cell culture, such as immunostaining or western blot of OC cell-specific marker genes.

2. The author sampled peripheral blood mononuclear cells (PBMCs) for in vitro induction to OC-like cell. What was the cell type component of PBMCs? Which type of cell in PBMCs was induced into OC-like cell?

3. Line 148, the full name of the abbreviation, DEG, should be labeled when it first appeared in the manuscript.

4. In Figure 1 b, which group was represented as PBMC and OC groups, respectively? 

Author Response

  • In the experiment design, the author used two factors, M-CSF and RANKL, to generate OC-like cells for 12 days of cultures. Is there any literature to support this cultural method? If not, the experiment is needed to prove the success of OC-like cell culture, such as immunostaining or western blot of OC cell-specific marker genes.

We have published extensively using OC-like cells generated with this protocol (see Mullin et al. 2014, 2018, 2019, 2020) and have confirmed their osteoclastic nature by staining for the osteoclast marker tartrate-resistant acid phosphatase (TRAP), capacity for resorption of bone and expression levels of established osteoclast marker genes. This has been added to the “Subject Recruitment and Generation of Osteoclast-like Cells” section of the Materials and Methods.

  • The author sampled peripheral blood mononuclear cells (PBMCs) for in vitro induction to OC-like cell. What was the cell type component of PBMCs? Which type of cell in PBMCs was induced into OC-like cell?

A description of the isolated PBMCs and the osteoclast precursor cells has been added to the “Subject Recruitment and Generation of Osteoclast-like Cells” section of the Materials and Methods. We have also added expression data for 3 monocyte marker genes to the “Analysis of Established OC Genes” section of the Results.

  • Line 148, the full name of the abbreviation, DEG, should be labelled when it first appeared in the manuscript.

The abbreviation “DEG” has now been defined when first used in the manuscript (see “Subject Recruitment and Generation of Osteoclast-like Cells” section of the Materials and Methods).

  • In Figure 1b, which group was represented as PBMC and OC groups, respectively?

Thank you, this has been corrected.

Round 2

Reviewer 1 Report

 1. I definitely know that use the same RNA that used for RNA-seq could get consistent result and there is no need to used the RNA that used for RNA-seq to do RT-qPCR. I mean that you need to use different batches of RNA to repeat the RNA-seq data to show the reproducibility of RNA-seq, which is some kind of varied between batches.

2. If you couldn't get new points or advance compared to the previous studies from the current study, then this manuscript is totally lack of interest to the potential readers.

3. By performing RNA-seq with only the starting and end point, you definitely get those DEGs, with two different samples, you will also get DEGs. At least several points is the common design to get a better understanding of a process from A to B cell type, especially, you need to publish the data that will finally be read by the potential readers. 

This revision obviously didn't address the key points. Based on the author's response, I couldn't provide any positive comments on this manuscript. If these points could not be addressed, I would suggest a rejection.

Author Response

Responses to reviewer comments

Reviewer 1

  • I definitely know that use the same RNA that used for RNA-seq could get consistent result and there is no need to used the RNA that used for RNA-seq to do RT-qPCR. I mean that you need to use different batches of RNA to repeat the RNA-seq data to show the reproducibility of RNA-seq, which is some kind of varied between batches.

Thank you for the clarification on this comment. We feel that validating the RNA-seq results using RT-qPCR is unnecessary as RNA-seq is considered more accurate than RT-qPCR; it gives an exact copy number of transcripts relative to total mRNA copy number, whereas RT-qPCR typically measures the relative difference in mRNA expression levels. RT-qPCR remains an important validation method for array-based RNA measurements, but we did not use those in this research. We have successfully published manuscripts previously containing RNA-seq data without RT-qPCR validation, including in Genes (PMID 34440306). A paragraph has been added to the Discussion considering this point.

  • If you couldn't get new points or advance compared to the previous studies from the current study, then this manuscript is totally lack of interest to the potential readers.

We respectfully disagree with the reviewer on this point. In this study, we have identified a large number of genes and biological pathways potentially involved in osteoclastogenesis in human cells, many of which are novel – the full list of the many interesting and important genes highlighted in the research is presented in Table S1. This will be of significant interest to researchers working in the bone field, especially considering that the osteoclast has proven to be such a successful target cell for pharmacological therapies aimed at treating various bone diseases to date (eg. Use of bisphosphonates for the treatment of osteoporosis, Paget’s disease and various bone malignancies). In response to the reviewer’s initial comment on this matter, we added additional information to the Results section of the manuscript on novel biological pathways identified in our analysis that appear related to osteoclast function. We feel that the comment has been adequately addressed.

  • By performing RNA-seq with only the starting and end point, you definitely get those DEGs, with two different samples, you will also get DEGs. At least several points is the common design to get a better understanding of a process from A to B cell type, especially, you need to publish the data that will finally be read by the potential readers.

We agree with the reviewer that a longitudinal study like the one described would certainly generate additional interesting results. However, this is beyond the scope of our cross-sectional study, which was aimed at identifying the main differences in gene expression driving osteoclastogenesis in our well-established cell culture model. We feel that we have successfully achieved this, and thereby provided over 3,000 gene identities for the focus of future research in the field. We may undertake such a longitudinal data collection in the future as a separate study, however, at present we have more than enough genes for downstream analysis and study.

Reviewer 2 Report

In the revised manuscript, the authors have addressed all my concerns. It can be published as it.

Author Response

Reviewer 2 does not need any further comments. He seems satisfied with our last revision.

Thank you.